# Characterisation of Antennal Sensilla and Electroantennography Responses of the Dung Beetles *Bubas bison*, *Onitis aygulus* and *Geotrupes spiniger* (Coleoptera: Scarabaeoidea) to Dung Volatile Organic Compounds

**DOI:** 10.3390/insects14070627

**Published:** 2023-07-12

**Authors:** Nisansala N. Perera, Russell A. Barrow, Paul A. Weston, Vivien Rolland, Philip Hands, Saliya Gurusinghe, Leslie A. Weston, Geoff M. Gurr

**Affiliations:** 1Gulbali Institute of Agriculture, Water and Environment, Charles Sturt University, Wagga Wagga, NSW 2678, Australia; nperera@csu.edu.au (N.N.P.); rubarrow@csu.edu.au (R.A.B.); pweston@csu.edu.au (P.A.W.); sgurusinghe@csu.edu.au (S.G.); leweston@csu.edu.au (L.A.W.); 2School of Agriculture, Environment and Veterinary Sciences, Charles Sturt University, Wagga Wagga, NSW 2678, Australia; 3CSIRO, Agriculture and Food, Canberra, ACT 2601, Australia; vivien.rolland@csiro.au (V.R.); philip.hands@csiro.au (P.H.); 4School of Agriculture, Environment and Veterinary Sciences, Charles Sturt University, Leeds Parade, Orange, NSW 2800, Australia

**Keywords:** olfaction, attractant, insect, basiconica, chaetica, trichodea

## Abstract

**Simple Summary:**

Insects, including dung beetles, rely on volatile cues to locate food and mates. However, the antennal responses of dung beetles to dung headspace volatiles have received minimal attention. To address this gap, we conducted a scanning electron microscopy study to examine the density distribution of three types of antennal sensilla in three introduced dung beetle species found in Australia: *Geotrupes spiniger*, *Bubas bison* and *Onitis aygulus*. The gross morphology of the antennal sensilla of these species is described here for the first time. Notably, we observed distinct patterns of sensilla trichodea, sensilla basiconica and sensilla chaetica on the proximal and distal surfaces of three lamellae in their antennal clubs. Furthermore, using electroantennography, we investigated the olfactory responses of these dung beetles to ten selected dung volatiles and mixtures of the same volatiles. The test chemicals evoked differential antennal responses in all three test species. The results are discussed in relation to the distribution and density of the antennal sensilla and the potential role of dung headspace volatiles in dung preference by these dung beetles. Overall, our findings indicate the possibility of using EAG-active compounds to attract dung beetles in the field.

**Abstract:**

Locating sporadically distributed food resources and mate finding are strongly aided by volatile cues for most insects, including dung beetles. However, there is limited information on the olfactory ecology of dung beetles. We conducted a scanning electron microscopy study on the morphology and distribution of the antennal sensilla of three introduced dung beetle species in Australia: *Geotrupes spiniger* (Coleoptera: Geotrupidae), *Bubas bison* and *Onitis aygulus* (Coleoptera: Scarabaeidae). Three main morphological types of antennal sensilla were identified: sensilla trichodea (ST), sensilla basiconica (SB) and sensilla chaetica (SCh). Distinct variations of SB distribution were observed in *B. bison* and *G. spiniger* and on different lamellar surfaces in both sexes of all three species. Sexual dimorphism in antennal sensilla distribution or their abundance was not evident. To complement the morphological characterisation of sensilla, electroantennography (EAG) was carried out to construct EAG response profiles of the three species to selected dung volatiles. An initial study revealed that antennae of all species were sensitive to a mix of phenol, skatole, indole, *p*-cresol, butanone and butyric acid, common components of livestock dung headspace. In addition to these six compounds, dimethyl sulfide, dimethyl disulfide, eucalyptol and toluene were tested for antennal activity. All compounds evoked measurable EAG responses, confirming antennal sensitivity. *Geotrupes spiniger* exhibited significant responses to all the compounds compared to the control, whereas *B. bison* and *O. aygulus* only responded to a subset of compounds. A comparison of relative EAG amplitudes revealed highly significant responses to *p*-cresol in *G. spiniger* and to skatole in *B. bison*. *Geotrupes spiniger* displayed differential responses to all the compounds. Pooled EAG data suggest highly significant differences in responses among the three species and among compounds. Our findings suggest that a blend of volatiles may offer potential for the trapping of dung beetles, thereby avoiding the use of dung baits that are inconvenient, inconsistent and may pose a threat to farm biosecurity.

## 1. Introduction

Dung beetles (Coleoptera: Scarabaeidae, Geotrupidae) are known for their significant role in the processing of animal dung, contributing to resource recycling in natural ecosystems especially in grazing environments. These insects mainly use vertebrate dung as their feeding and breeding source, resulting in dung decomposition, nutrient recycling, improved soil fertility and soil aeration, reduced greenhouse gas emissions and the suppression of parasites and flies [1,2,3,4,5]. As in the case of many other insects, dung beetles respond to odours associated with feeding and breeding sources. Therefore, their attraction to dung is induced by volatile organic compounds (VOCs) emitted by dung pats and conspecific adult beetles, respectively [6,7,8,9]. Beetles move upwind to locate fresh dung pats by perceiving volatile compounds emitted from the dung [10]. Based on studies performed on other insects, odorant-guided food and mate navigation is achieved via olfactory receptor neurons (ORNs) housed in the olfactory sensilla that cover the surface of the antenna [11,12]. Once odorant molecules reach a sensillum, they bind with odorant-binding proteins (OBPs) and the complex is transported through the sensilla lymph to activate the olfactory receptors (ORs) [11,13,14,15]. However, a previous study has also demonstrated that odorant transportation can occur in the absence of OBPs in *Drosophila* [16]. 

In insects, antennae are the primary sensory organ and house different types of sensilla classified primarily on their external morphology [17,18], performing various functions such as olfaction, gustation, mechanical reception, thermoreception and hygroreception [19,20,21,22,23]. The morphology and function of an antenna can be defined by the ecological niche occupied; this niche is subjected to selective pressure, influencing chemical communication and thereby increasing the signal-perceiving efficiency [24,25]. Therefore, sensilla type, density and distribution on antennae may impact the sensitivity profile for odorant molecules. Although the antennal morphology and sensilla structure have been characterised in many coleopteran families, the antennal sensory apparatus of dung beetles is still poorly understood [26,27]. Only a few studies have investigated the morphology and distribution of antennal sensilla in dung beetles, specifically *Geotrupes auratus*, *Copris pecuarius* [28], *Typhaeus typhous*, *Onthophagus fracticornis* and *Aphodius fossor* [29]. 

The types and levels of odorants that insects can detect and how they perceive information is typically species-specific [30,31,32,33,34]. A single VOC can trigger a behavioural response in some insects, and olfactory studies are necessary to determine whether a chemical stimulus can be translated into an antennal response. Recently, it has been shown that dung beetle olfactory systems can recognize and discriminate amongst various VOCs in the environment. For example, several studies have reported variation in attraction and clear trophic preferences to dung by dung beetles under a range of field and laboratory conditions [6,33,35,36,37,38,39]. Sladecek et al. [40] present empirical evidence as to how the temporal variation of volatiles emitted from cow dung pats influences the community dynamics of dung inhabitants, mainly beetles and flies. Qualitative and quantitative composition of dung volatilomes vary with time [41], the diet of the host animal [38], the gut microbial fauna of the host animal and the activity of soil microbes [40,42]. Selection pressure on the insect may have resulted in species-specific olfactory mechanisms by which different insects process information. Therefore, the potential of a compound to signal as a background odour or a resource-indicating odour and the distribution and abundance of olfactory receptors may vary based on the beetle species and their ecological niche [27,29,43,44,45].

Behavioural studies have shown that when dung beetles orient to a dung source, they are more likely to utilise specific blends of VOCs rather than a single compound, suggesting a synergistic effect among volatiles [38,46,47,48]. Studies have shown that different host diet regime, sex or the life stage of the host animal may produce a distinct dung volatile profile which can have an impact on the dung beetle species attracted [38,42]. Although the EAG activity of dung beetles in response to dung volatiles has received minimal attention [41,48,49], there has been a comprehensive set of studies performed that characterise the antennal responses of the dung beetle *Kheper* spp. to semiochemicals [50]. In addition, single sensillum recordings (SSR) revealed that a Japanese dung beetle, *Geotrupes auratus*, has two specific clusters of olfactory cells: R-type I, which responds to butanone, and R-type II, which responds to several other compound cues, including butanone, *p*-cresol, indole, phenol and skatole [48]. Urrutia et al. [49] demonstrated that certain EAG-active compounds including *p*-cresol and skatole can influence the feeding preference of some dung beetle species.

The number of studies performed on chemical communication and the role of olfactory receptors in detecting odour cues in dung beetles is minimal. Our previous study showed that the generalist dung beetle *Bubas bison* can orient towards a specific odour bouquet, proving the involvement of certain dung headspace volatiles in dung attractancy [38]. That study also provided evidence that *B. bison* was especially attracted to horse dung that characteristically emits skatole, indole, *p*-cresol, phenol, butyric acid, toluene, butanone, dimethyl sulphide, dimethyl disulphide and eucalyptol. Based on those findings, we selected this group of compounds as potential beetle attractants for further studies. Indole, skatole, phenol and butyric acid have been described as dung beetle pheromone constituents (Table 1) in the genus *Kheper*. Furthermore, some of these compounds have recently been tested in the field in various combinations as dung beetle attractants, especially in Europe [46,47,51]. However, the relative attractancy of dung VOCs to different dung beetle species remains unclear, especially with regard to their species-specific role. In general, compounds have shown greater attractancy when included as a part of a blend rather than tested individually, suggesting a synergistic effect from multiple constituents [38,46,47]. 

Given the paucity of data available, we undertook a series of studies to characterise and quantify the dung beetle antennal sensilla in male and female adult dung beetles and to screen selected dung VOCs for EAG activity as an indication of potential for chemical attractancy in the field. We hypothesize that the olfactory responses of dung beetles to dung VOCs may be associated with the distribution of various antennal sensilla. Therefore, comparisons were made across three dung beetle species representing two coleopteran families: *Geotrupes spiniger* (Coleoptera: Geotrupidae), *Onitis aygulus* and *Bubas bison* (Coleoptera: Scarabaeidae). *Geotrupes spiniger* was first introduced to Australia in 1979, *O. aygulus* in 1977 and *B. bison* in 1983 [52] as part of the Australian Dung Beetle Project. These beetles now play a crucial role in dung burial in Australian pastures. Here we describe the distribution and the density of various types of antennal sensilla in *G. spiniger*, *O. aygulus* and *B. bison* for the first time. Additionally, the olfactory receptor sensitivity of *O. aygulus* and *G. spiniger* is documented for the first time. We establish the presence of chemoreceptors on the antennal club of all three species for the candidate chemicals, despite the diverse range of compounds. Our ultimate goal is to determine the ecological role and potential of EAG-active dung VOCs for use as field lure by elaborating their ability to act as either a resource-indicating odour or a background odour for dung beetles. This approach will further direct the development of dung volatile-based chemical lures that could potentially replace dung baits in the future, as dung bait efficacy is inconsistent and temporal in nature and may pose a threat to on-farm biosecurity through the unintentional spread of pathogens present in dung.

**Table 1 insects-14-00627-t001:** Occurrence of VOCs used in the current study in different livestock dung types and dung-mimicking organisms. Additionally, their role as semiochemicals and existing information on the potential EAG activity for dung beetle species. *Bubas bison* has been tested for indole, skatole, *p*-cresol and eucalyptol. No EAG- activity data can be found for *O. aygulus* and *G. spiniger* for any of the compounds tested.

Compound	Structure	Compound Group	Presence in Livestock Dung	Presence in Dung Mimicking Organisms	Presence as a Dung Beetle Semiochemicals	Antennal Responses by Dung Beetle Species
indole	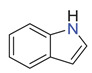	indoles and derivatives	Cow [49]Cow, horse, sheep and boar [9]Fox [46]Horse and sheep [38]Cow [42]	*Wurmbea elatior* [53] *Typhonium brownie* and *T. eliosurum* [54] *Arum* spp. [55]	Male abdominal secretions of *Kheper bonellii* [50]	*G. auratus* [48]*Onthophagus binodis* [41]*Ammoecius elevates*, *Anomius baeticus*, *Aphodius fimetarius*, *Ceratophyus hoffmannseggi*, *Jekelius hernandezi*, *Sericotrupes niger*, *Thorectes valencianus*, *Typhaeus typhoeus*, *Ateuchetus cicatricosus*, *Bubas bison*, *Copris hispanus*, *O. emarginatus*, *O. fracticornis*, *O. maki* and *O. melitaeus* [49]
skatole	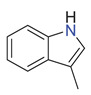	indoles and derivatives	Cow, horse, sheep and boar [9]Horse [49]Cow, horse and sheep [38]Cow [42]	*W. elatior* [53] *T. brownie* and *T. eliosurum* [54]	Male abdominal secretions of *K. lamarchi*, *K. nigroaeneu*, *K. subaeneus* and *K. bonellii* [50]	*G. auratus* [48]*Ammoecius elevates*, *Anomius baeticus*, *Aphodius fimetarius*, *Ceratophyus hoffmannseggi*, *Jekelius hernandezi*, *Sericotrupes niger*, *Thorectes valencianus*, *Typhaeus typhoeus*, *Ateuchetus cicatricosus*, *Bubas bison*, *Copris hispanus*, *O. emarginatus*, *O. fracticornis*, *O. maki* and *O. melitaeus* [49]
phenol	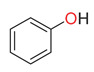	phenolic compounds	Cow, fox [46]Cow, horse and sheep [38]Weka [7]Cow [42]		Pygidial gland secretions of *Canthon cyanellus cyanellus* and *C. femoralis femoralis* [56]	*G. auratus* [48]*O. binodis* [41]
*p*-cresol	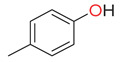	phenolic compounds	Cow and horse [49]Cow, horse, sheep and boar [9]Horse, sheep, deer, cow, fox, and wild boar [46]Cow, horse and sheep [38]Cow [42]Weka [7]	*Typhonium brownie* and *T. eliosurum* [54] *Arum* spp. [55]		*G. auratus* [48]*Onthophagus binodis* [41]*Ammoecius elevates*, *Anomius baeticus*, *Aphodius fimetarius*, *Ceratophyus hoffmannseggi*, *Jekelius hernandezi*, *Sericotrupes niger*, *Thorectes valencianus*, *Typhaeus typhoeus*, *Ateuchetus cicatricosus*, *Bubas bison*, *Copris hispanus*, *O. emarginatus*, *O. fracticornis*, *O. maki* and *O. melitaeus* [49]
butanone	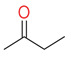	ketone	Cow, horse and sheep [38]Cow [42]			*G. auratus* [48]
butyric acid	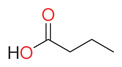	fatty acids and conjugates	Weka [7]Cow [42]		Male abdominal secretions of *K. subaeneus* and *K. bonellii* [50]	
eucalyptol	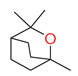	monoterpenes	Horse [38]Rabbit [49]			*O. binodis* [41]*Ammoecius elevates*, *Anomius baeticus*, *Aphodius fimetarius*, *Ceratophyus hoffmannseggi*, *Jekelius hernandezi*, *Sericotrupes niger*, *Thorectes valencianus*, *Typhaeus typhoeus*, *Ateuchetus cicatricosus*, *Bubas bison*, *Copris hispanus*, *O. emarginatus*, *O. fracticornis*, *O. maki* and *O. melitaeus* [49]
toluene	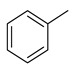	benzene and substituted derivatives	Cow, horse and sheep [38]Cow [42]			*Onthophagus binodis* [41]
dimethyl sulfide	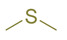	thioethers	Horse and sheep [38]Cow [42]			
dimethyl disulfide	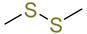	Aliphatic disulfides	Horse and sheep [38]Cow [42]			

## 2. Materials and Methods

### 2.1. Collection of Beetles 

Freshly eclosed adult *B. bison and O. aygulus* beetles used in the current study were collected from the Charles Sturt University farm, Wagga Wagga, New South Wales, and *G. spiniger* were collected from Hobart, Tasmania. Beetles were maintained in a glass house at 20–22 °C in 20-L (*B. bison* and *O. aygulus*) and 100-L (*G. spiniger*) containers filled with moist vermiculite and provisioned with cattle dung as the food source until use in the experiments. 

### 2.2. Scanning Electron Microscopy (SEM) of Antennal Sensilla

Male and female antennae were carefully excised from the live adult beetles (*n* = 3) and dehydrated by immersing them in an ascending series of ethanol solutions (50%, 70%, 90% and 100%) for five minutes in each solution. Samples were gently cleaned in an ultrasonic water bath (Powersonic 410, Wetherill Park, NSW, Australia) followed by critical point drying (Tousimis 931, Tousimis Research Corp., Rockville, Maryland, USA). Whole antennae and dissected lamellae were mounted on aluminium stubs with double-sided adhesive carbon tabs (ProSciTech, Townsville City, QLD, Australia). Samples were gold coated using an Emitech K550X sputter coater (Quorum Emitech, Kent, UK) at 25 µA for 2 min. Samples were imaged using a Zeiss EVO LS 15 scanning electron microscope (Carl Zeiss Microscopy GmbH, Jena, Germany) with 20 kV accelerating voltage for whole antenna samples and 15 kV for individual lamella, at 10 Pa vacuum, a spot size of 500 and a backscattered electron detector. Olfactory receptors were identified and classified based on studies by Zacharuk (1985) [57], Steinbrecht (1996) [58], Schneider (1964) [18] and Zauli et al. [26]. Sensilla were enumerated on each lamella, over an area of 2500 µm^2^ (50 μm × 50 μm), defined as a region of interest (ROI). ROIs were selected (*n* = 5–12) to cover both surfaces of each lamella and care was taken not to avoid sloping areas. No depth correction was performed because there was no unbiased method available. The density of each sensilla type was calculated per 100 µm^2^ [28].

### 2.3. Chemicals and Odour Stimuli Preparation

The sensitivity of female dung beetle antennae to ten dung VOCs was assayed by EAG. Candidate compounds, which included aromatic compounds (skatole, indole, *p*-cresol, phenol and toluene), aliphatic compounds (butyric acid, butanone, dimethyl sulphide and dimethyl disulphide) and one monoterpene (eucalyptol), were selected based on their importance in dung beetle attraction as reported by other researchers [9,46,47,59] and our previous study on *B. bison* [38]. Commercially available standards were purchased from Sigma-Aldrich, Australia (Appendix A) and diluted in laboratory grade water [48] to a concentration of 10 µg/mL, as optimized during preliminary experiments. A mix of six compounds consisting of skatole, indole, *p*-cresol, phenol, butyric acid and butanone was used to assure successful antennal dissection and preparation. A Pasteur pipette was used to load each odour sample and when not in use, the two ends of the pipette were covered with Parafilm to reduce the vaporisation of the stimulus solution. 

### 2.4. Antennal Preparation and Electroantennogram Recordings

The EAG technique used in this study was similar to that previously described in the literature [60,61,62] and recordings were conducted only with female antennae (*n* = 3–11). Adult beetles were starved overnight and kept at 4 °C for 15 min to reduce their activity before dissection. Antennae were carefully excised from the base under a stereomicroscope (Nikon C-LEDS, Nikon Corporation, Shanghai, China) with the aid of a sharp scalpel and a probe. The antennae were immediately mounted between reference and recording glass electrodes containing an electrode gel (Spectra 360 Electrode Gel, Parker Laboratories, Fairfield, USA). The scape and pedicel were either removed or inserted into the glass electrode to avoid possible noise generated by the mechanoreceptors (mainly sensilla chaetica) present on those segments. A short length of human hair was used to separate the lamellae of the antennal clubs in order to provide maximum exposure of the olfactory receptors which line the interior surfaces of the lamellae [60]. The tip of a Pasteur pipette containing a piece of folded filter paper (Whatman #1, 1 × 2 cm) loaded with 20 µL of test solution was inserted into the small hole of the mixing tube of the EAD apparatus. Compounds belonging to the same chemical group were assumed to interact with the filter paper and desorbed into the headspace in an equal manner. A continuous flow of charcoal-filtered, humidified air was delivered through the mixing tube at 400 mL/min using a Syntech stimulus controller (CS-55, Syntech, Kirchzarten, Germany), to which a stimulus puff was introduced for 0.5 s at 1 L/min via the Pasteur pipette. Once a stable baseline was achieved, a puff of the six-compound mix was used to confirm antennal activity. Three consecutive puffs of each test compound were applied at 30-s intervals, after which the antennal preparation was flushed with clean air for ca. 15 min. The recovery of the antennae was confirmed by applying a puff of control stimulus (water). The testing sequence of these compounds was randomised for each species. For each test species, compounds were tested on 3–11 antennae excised from different individuals. For each stimulus, the maximum amplitude (mV) was recorded. 

### 2.5. Data Analysis

SEM datasets were checked for normality using the Shapiro–Wilk test (*p* > 0.05). The length of various antennal segments was measured using ImageJ software (v.1.53t, National Institutes of Health, Bethesda, MD, USA) and reported as mean ± standard error. Data on antennal segment length for males and females of each species were analysed using Student’s *t*-tests (α = 0.05) [63]. The density and distribution of sensilla were quantified for sensilla trichodea (ST), sensilla basiconica (SB) and sensilla chaetica (SCh), and significant differences in distribution among lamellar surfaces were compared using a one-way analysis of variance (ANOVA) followed by Tukey’s HSD multiple comparison (α = 0.05). Given their importance in olfaction, the density of SB and ST was pooled and analysed using an ANOVA. Additionally, for males and females, a factorial ANOVA was used to determine the effect of species, sensilla type and lamella surface on sensilla density. All analyses were performed using Statistix software (ver. 10, Analytical Software, Tallahassee, FL, USA) and SPSS (ver. 29). 

EAG datasets were checked for normality using the Shapiro–Wilk test (*p* > 0.05). Mean values for three consecutive puffs from each compound for the same antennae were calculated. A paired *t*-test was used to compare responses between the compound and the respective control for each species. Then, EAG responses for the most recent control were subtracted from absolute EAG values for the test compounds to compensate for the consecutive control [49]. A one-way ANOVA was performed followed by either Fisher’s least significant difference (LSD) test [64,65] or Tukey’s significant difference test when LSD was unable to detect the significant differences due to statistical limitations (*p* < 0.05). Welch’s Test for Mean Differences [66] was used for mean comparison when homogeneity of variances was violated as determined by Levene’s test [67]. For each species and compound, mean relative responses were pooled and compared with ANOVA followed by LSD [68]. All analyses were performed using Statistix software (ver. 10, Analytical Software, Tallahassee, FL, USA). 

## 3. Results

### 3.1. Comparative Morphology and the Variation of Antennal Sensilla

#### 3.1.1. General Morphology of the Antennae

The antennae of all three species showed a typical lamellicorn shape consisting of scape, pedicel, segmented funicle and antennal club with three lamellae (L) (Figure 1 and Figure 2a). The scape and pedicel were singly segmented in all three species. Both *B. bison* and *O. aygulus* possessed four flagellomeres (F) in the funicle, but *G. spiniger* had six flagellomeres. Except in the F1 and F2 in *O. aygulus*, no significant differences were found between males and females in the lengths of the antennal segments in all species. Male *O. aygulus* had significantly longer F1 and F2 compared to conspecific females (Appendix A). Furthermore, in both species of scarabaeid beetles, flagellomere length decreased from F1 to F4, towards lamellae, but in geotrupid, F3 was longer than F2 and F4. In all three species examined, the last three antennal segments were modified into three lamellae that could be folded into a club. The proximal surfaces of all lamellae were convex, while the distal surfaces were concave for L1 and L2 but took on a flattened appearance for L3. The shape of L1 and L2 lamellae in the geotrupid *G. spiniger* was less arcuate compared to that of scarabs, *B. bison* and *O. aygulus* (Figure 1). 

#### 3.1.2. Main Antennal Sensilla Types and Their Distribution on Each Lamella

Three distinct types of sensilla were observed on the antennae of the beetles examined: sensilla chaetica (SCh), two subtypes of sensilla trichodea (ST) and two subtypes of sensilla basiconica (SB). Cuticular pores on the surface of the lamella were also identified. Sensilla chaetica were characterised by a distinct circular membrane at the base and a pointed tip and were immersed in a deep socket. Furthermore, compared to ST, SCh were both longer and had a wider base (Figure 2b). Sensilla trichodea were long, slender and hair-like without a specialized basal membrane. Two subtypes of ST were recognised based on their external appearance; ST I was short with a slightly curved and flattened tapering end while ST II was longer, thicker and more sharply tipped than ST I (Figure 2b). The majority of sensilla trichodea across all species examined were ST I and were most abundant on the protected inside surfaces of lamellae, while ST II were confined to outside surfaces. Sensilla basiconica were conical with a blunt tip. Two morphological subtypes were identified; SB I was short, arising from a shallow pit, whereas SB II was longer and slightly curved towards the apex compared to SB I (Figure 2b). Data for the density of SCo are not shown here as they were distributed sporadically at very low densities compared to SCh, ST and SB; therefore, quantification was difficult. Here we focused on three major sensilla types which were highly abundant for quantification (ST, SB and SCh) considering their function and importance as insect chemoreceptors. 

***Bubas bison*:** SEM images revealed the presence of all three major types (including all subtypes) of sensilla on both proximal and distal surfaces of all three lamellae of both sexes of *B. bison*. When comparing male beetles to female beetles, a significantly higher density of ST was observed on L3 proximal surfaces while a higher density of SB was observed on both L1 and L3 proximal surfaces (Table 2). Sensilla basiconica was the most abundant type of sensilla across all surfaces of the antennal club and was dominated by SB II. On the distal surfaces of L1 and L2 lamellae, two clear zones were identified: a homogenous area next to the flagellum consisting of SB II and an outer heterogenous area containing a mixture of SB II and ST I (Figure 3). Sensilla trichodea was dominated by ST I on almost all the lamellar surfaces except the L1 proximal surface, where ST II were more abundant. 

***Onitis aygulus*:** Sensilla types present in *O. aygulus* were similar to those of *B. bison*. SB I and SB II were intermingled and distributed across all lamellar surfaces, but no distinguished areas of SB were observed. Sensilla basiconica was dominated by SB II across all lamellae. The densities of ST and SB on L3 proximal surface of male beetles were significantly greater than on those observed for females (Table 2). 

***Geotrupes spiniger*:** All three main sensilla types, namely SCh, ST and SB, were observed on the antennal clubs of *G. spiniger*. The density of SB observed on the L2 proximal surface in male beetles was significantly higher than in females. Interestingly, as in *B. bison*, the distal surfaces of L1 and L2 showed two distinct zones: a homogenous inner zone consisting of densely packed SB I and an outer heterogenous zone with mixed SB I, SB II and ST I (Figure 3). Furthermore, the density of ST and SB on L1 and L2 distal surfaces were similar in both male and female *G. spiniger* and *B. bison* (Figure 4). Differentiating L1 lamella of *G. spiniger* from *B. bison* was the fact that the L1 proximal surface of *G. spiniger* was covered by a cuticular plate where only cuticular pores existed (Figure 2a). 

In all three species, SCh were mainly distributed on external surfaces of the antennal club; L1 proximal and L3 distal, with some on the peripheral edges of every lamella, mixed with other sensilla. The density of SCh on the L1 proximal surface was significantly greater than on the L3 distal surface (Table 2). The density and distribution of ST, SB and SCh among each lamellar surface in the three dung beetle species studied were significantly different, with the exception being for the L2 distal surface in *O. aygulus* (Figure 4). The distribution pattern across lamellar surfaces and pooled data within each species suggest similar ST and SB distribution in *B. bison* and *G. spiniger*. L1 and L2 distal lamellar surfaces had higher densities of ST and SB compared to the proximal surfaces in both males and females (Figure 4, Appendix A). On the other hand, in *O. aygulus* ST and SB had a similar distribution density on both distal and proximal lamellar surfaces. The L1 proximal surface in all three species had the overall lowest sensilla distribution (Figure 4 and Appendix A). Moreover, SB can be identified as the most abundant sensilla type in both males and females. Multifactorial analysis revealed a strong statistically significant effect of lamella surface and sensilla type as well as two- and three-way interactions among lamella surface, species and sensilla type on the sensilla density, as indicated by the *p* values (<0.001) (Table 3). However, neither sex nor the species seemed to influence the sensilla distribution except the two-way interaction between sex and sensilla type (*p* = 0.031). The model was a good fit, as indicated by the *R* squared value (*R*^2^ = 0.968). 

### 3.2. EAG Responses of Female Beetles to Selected Dung VOCs

All test compounds evoked reproducible EAG responses in all three species when presented separately and in the six-compound mix, which confirmed antennal activity. Freshly dissected antennae could be used for 4–5 h without a noticeable decrease in EAG response for all three species. When compared with the respective control, female B. bison significantly responded to the six-compound mix, skatole, butyric acid, DMS, toluene, *p*-cresol and phenol, whereas female *O. aygulus* significantly responded to the six-compound mix, skatole, eucalyptol, butyric acid, toluene, *p*-cresol and phenol. In contrast, the antenna of female *G. spiniger* responded significantly to all test compounds and the six-compound mix. When presented as radar charts, the highest overall antennal response occurred in *B. bison* and the lowest in *O. aygulus* (Figure 5, Appendix A). The EAG amplitudes for *B. bison* ranged from 0.556 ± 0.122 (skatole) to 0.269 ± 0.087 (DMDS); for *O. aygulus* responses ranged from 0.395 ± 0.057 (toluene) to 0.107 ± 0.017 (eucalyptol) and for *G. spiniger* the range was from 0.660 ± 0.049 (*p*-cresol) to 0. 211 ± 0.047 (DMS) (Appendix A). Considering absolute EAG responses among the three species, skatole (*F*_2,8.6_ = 5.52, *p* = 0.0288) and *p*-cresol (*F*_2,9_ = 7.73, *p* = 0.011) showed statistically significant EAG amplitudes (Figure 6, Appendix A). An LSD pairwise comparison showed that skatole and *p*-cresol evoked the highest EAG signals in *B. bison* and *G. spiniger*, respectively (Figure 6). 

Relative mean EAG amplitude comparisons within species showed that EAG responses for individual dung VOCs were statistically significant only for *G. spiniger* (*F*_10,51_ = 6.49, *p* < 0.001). A subset of EAG-active compounds elicited the highest responses; *p*-cresol was highest, followed by the mix, skatole, indole, phenol, butyric acid and DMDS, whereas butanone, eucalyptol, DMS and toluene elicited the lowest signals in *G. spiniger* antennae (Figure 7a). A radar plot shows different EAG sensitivity profiles for all test compounds for the three species (Figure 7b). However, in *O. aygulus,* except for DMS, butanone, toluene and phenol, all other test compounds showed a lower response when compared to *B. bison* and *G. spiniger*. Factorial analysis revealed a significant effect of compound (*F* = 2.724, *p* = 0.005, *df* = 10) and species (*F* = 6.615, *p* = 0.002, *df* = 2) on EAG responses, although the compound × species effect was not statistically significant (*F* = 0.942, *p* = 0.536, *df* = 20) (Table 4). Pooled EAG responses were statistically significant among species (*F*_2,158_ = 9.37, *p* = 0.0002). Overall, the pooled EAG amplitude was highest in *B. bison* (0.397 ± 0.026) and lowest in *O. aygulus* (0.235 ± 0.032) (Table 5). When EAG responses were pooled per species, the mean responses were statistically different among compounds (*F*_10,150_ = 2.61, *p* = 0.0059). Of the ten individual compounds and the mix, *p*-cresol elicited the highest response from beetle antennae (0.450 ± 0.048) compared to butanone, eucalyptol and DMDS (Table 6). 

## 4. Discussion

The density and distribution of three major antennal sensilla in three dung beetle species, representing two families, along with the EAG activity of female beetles in response to ten dung VOCs were studied. Our results show an association between the distribution of antennal sensilla on the antennae and the EAG response to dung VOCs in the studied species. All three species have segmented lamellicorn antennae [69]. We also report for the first time on the identity and distribution of three types of sensilla on the antennal clubs of *B. bison*, *O. aygulus* and *G. spiniger*. Sensilla chaetica (SCh), sensilla basiconica (sub types; SB I and SB II) and sensilla trichodea (subtypes; ST I and ST II) were identified in all three species and were differentially distributed. The sensilla identified are similar to those previously described in the dung beetles *Aphodius fossor* and *Typhaeus typhoeus*, *Onthophagus fracticornis* and *Geotrupes auratus* [28,29].

Sensilla basiconica were the most abundant sensilla type across all lamellar surfaces of the three species. Several studies have also reported on the olfactory function of sensilla basiconica in the perception of various chemical cues, including host-associated volatiles and potential sex pheromones in insects [70,71,72,73]. Among different lamellar surfaces, the density of SB on the distal surfaces of L1 and L2 in *B. bison* (SB II) and *G. spiniger* (SB I) were significantly greater than the density of ST. Densely clustered homogenous areas of SB may indicate highly sensitive odour-perceiving areas on distal surfaces of L1 and L2 in both species. A similar spatial separation of lamellar surfaces was observed in some scarabs with sensilla basiconica [29] and sensilla placodea [26,74,75]. SB I were similar in morphology to those described in congeneric *G. auratus* and SB II were similar to sensilla basiconica found in *Phoracantha semipunctata* (Coleoptera: Cerambycidae) [71] and the dung beetles *C. pecuarius* and *O. fracticornis* [28,29]. Sensilla basiconica in *G. auratus* are also densely arranged, and each sensilla contains two olfactory cells [28,48]. However, in *O. aygulus* no such differences were observed in relation to SB in distal lamellar surfaces; rather, a similar abundance was observed for both ST and SB. 

Trichoid sensilla were the second most abundant type of sensilla detected in this study. ST occurred on all lamellar surfaces in both sexes of the three test species, and a relatively higher density was observed on proximal lamellar surfaces in all three species. The olfactory function of innervated ST has been reported in many insects [12,20,21,72,73,76,77,78]. Both types of ST identified in our study are morphologically similar to sensilla trichodea found in the dung beetles *O. fracticornis*, *A. fossor* and *T. typhoeus* [29]. We found that sensilla trichodea in *O. aygulus* did not display a significant density difference with the distribution of SB, which contrasts with our observations in *B. bison* and *G. spiniger*. When compared to other Scarabaeoidea families, dung beetles in the families Scarabaeidae and Geotrupidae have been found to have the highest density of ST and SB, as reported by Bohacz et al. [29]. As for sensilla chaetica, we found that the L1 proximal surface had a higher density than the L3 distal surface in all three species. Electrophysiology experiments have shown that SCh has both mechanoreception and gustatory functions [19,79,80]. Given the presence of SCh on only external surfaces of the antennal club (L1 proximal and L3 distal), it may suggest either a mechanical or gustatory (non-olfactory) function in the dung beetles evaluated in this study. The literature suggests a conserved olfactory system between congeneric [75,81,82,83] and other species [84]. As dung beetles share a similar feeding strategy and occupy a specific ecological niche, it might be expected that a similar sensilla arrangement would be maintained throughout a family. However, it is unclear why *B. bison* and *G. spiniger*, which belong to different families (Scarabaeidae and Geotrupidae, respectively), have similar ST and SB distributions on their lamellae. It is also uncertain why *O. aygulus*, which belongs to the same family as *B. bison*, does not share a similar sensilla arrangement.

Although recent studies tested the antennal sensitivity of *B. bison* to several dung VOCs [38,49], we document the EAG activity of *G. spiniger* and *O. aygulus* here for the first time, and we also expand the knowledge of EAG activity for *B. bison*. As reflected in Table 1, many of the compounds we tested have been detected in dung headspace and as constituents in dung beetle semiochemicals. Moreover, compounds that were in the six-compound mix (skatole, butyric acid, indole *p*-cresol, butanone and phenol) have been tested in previous studies to determine their field attractiveness to dung beetles. Our results reveal that selected compounds belonging to diverse chemical groups were able to elicit olfactory responses in the antennae of female dung beetles compared to the control, confirming the presence of olfactory receptors. Statistically significant and consistent EAG responses were observed against the mix of six compounds, skatole, butyric acid, toluene, *p*-cresol and phenol across all three species, suggesting the potential role of these compounds in dung attractiveness. Fluctuations in EAGs were observed because the recording included responses from a larger number of olfactory neurons acquired at once. Consequently, a differential degree of sensitivity for different stimuli can be expected. The largest EAG amplitudes were obtained for skatole, toluene and *p*-cresol when compared to the control for *B. bison*, *O. aygulus* and *G. spiniger*, respectively. This differential odour sensitivity could potentially be attributed to the differential distribution of olfactory receptors responding to chemical cues. As an example, the middle lamella of the scarab beetle *Pseudosymmachia flavescens* (Coleoptera: Scarabaeidae: Melolonthinae) generated higher EAG responses than those of the proximal or distal lamella and the closed antennal club, which was found to have a significant correlation with the density of sensilla placodea in a previous study [27,62]. Single sensillum recordings (SSRs) have shown that olfactory neurons housed in densely arranged sensilla basiconica in *G. auratus* were sensitive to indole, skatole, phenol and *p*-cresol, while a separate group of olfactory cells responded specifically to butanone [28,48]. Similarly, we propose that conserved areas of homogeneous SB in inner, protected surfaces of the lower and middle lamella in *B. bison* and *G. spiniger* may generate stronger EAG responses in those species compared to *O. aygulus*. Co-localisation of ST and SB in relatively lower densities may consequently produce an overall weaker EAG response in *O. aygulus*.

When comparing absolute EAG values, both skatole and butyric acid were able to evoke strong responses in *B. bison* antennae and weak responses in *O. aygulus* antennae. This is consistent with findings from a recent study, which found skatole and *p*-cresol to be EAG-active for *B. bison* [49]. Skatole has also been tested as a field bait with several other chemicals and has proven to be attractive for dung beetles [46,47]. Other insects have been found to possess olfactory receptors sensitive to skatole [85]. Our previous work revealed that *p*-cresol was attractive to *B. bison* in an olfactometer bioassay as a single compound [38], and here we show that *p*-cresol can also generate a strong response in the antennae of *G. spiniger* and to a lesser extent in *B. bison*. *Geotrupes spiniger* has been found to prefer cattle dung in the field [32], and recently *p*-cresol has been detected in cattle dung headspace [49]. Our findings also suggest that, for *G. spiniger*, *p*-cresol has a significant effect on eliciting an antennal response over other compounds tested. Such high sensitivity could be due to the ability of *G. spiniger* to detect *p*-cresol more effectively than other compounds, which indicates the role of *p*-cresol as a resource indicator volatile for *G. spiniger*, whereas other compounds may be acting as background volatiles. When comparing EAG values against the control, both *B. bison* and *O. aygulus* antennae produced weaker responses within the species than those in *G. spiniger*. The obvious difference between *G. spiniger* and *B. bison* and *O. aygulus* is the type of SB present. In *G. spiniger*, short basiconica sensilla, SB I, was observed, while in the scarabs the longer SB II were present. It is possible that the scarabs *B. bison* and *O. aygulus* may have receptors for these compounds on SB II that induce a generic response. Toluene in this case tends to elicit a relatively stronger response in the scarab beetles (*B. bison* and *O. aygulus*) while giving a weaker response in the geotrupid beetle (*G. spiniger*). Our previous work provides evidence for the enhanced attractancy of horse dung for *B. bison*, which contains a high proportion of toluene [38]. The quantities and ratios in the headspace that can stimulate olfactory receptors remain unknown to date. Therefore, future single sensillum recordings would help to determine specific responses to those headspace compounds. 

Pooled olfactory responses among the three species differed significantly, indicating potential species-specific sensitivity for dung VOCs. Notably, the summation of the sensilla basiconica density, either for L1 and L2 distal surfaces or for the whole antennal club, is greatest in *B. bison,* followed by *G. spiniger* and *O. aygulus,* which is consistent with previous findings. Antennal responses to the six-compounds mix, skatole, butyric acid and eucalyptol followed the same trend for the three species. Skatole was found to be detected by basiconica sensilla in dung beetles [48] and by trichoid sensilla in mosquitoes [86]. According to the pooled data for compounds, the largest EAG amplitude was obtained for *p*-cresol while the lowest responses were obtained for butanone, eucalyptol and DMDS. Aliphatic DMS and butyric acid, aromatic indole and phenol had similar responses. Dimethyl trisulfide (DMTS) was previously suggested to be a primary chemical cue for late-colonising, dung-inhabiting beetles [40]. It also stimulated strong responses in the antennae of *Anoplotrupes atercorosus* [87]. Wurmitzer et al. [47] showed the importance of butyric acid as an attractant for dung beetle communities. 

In conclusion, the morphology and distribution of three distinct antennal sensilla in *B. bison*, *O. aygulus* and *G. spiniger* were characterised. Sensilla basiconica may have a potential olfactory function, as determined by their distribution and EAG responses to dung VOCs. Sexual dimorphism was not observed in sensilla types or distribution (Appendix A). The significant EAG responses detected in the study species confirms the involvement of antennal receptors for dung volatile detection. The characterisation of antennal sensilla and data on antennal responses provide an important foundation for future studies on dung beetle chemical ecology. However, the behavioural significance of these sensilla associated with olfaction in dung beetles remains largely unexplored. Therefore, our findings point to the need for future investigation of the functional role of sensilla in dung volatile recognition using transmission electron microscopy coupled with single sensillum recordings. Furthermore, the volatile compounds that influence dung beetle olfactory preference behaviour remain to be characterised. Differences in sensilla distribution and abundance and as yet unknown environmental factors are likely to contribute to the behavioural responses of dung beetles to dung. At this time, field studies are in progress with the aim of formulating a field attractant for dung beetles using EAG-active compounds. 

## Figures and Tables

**Figure 1 insects-14-00627-f001:**
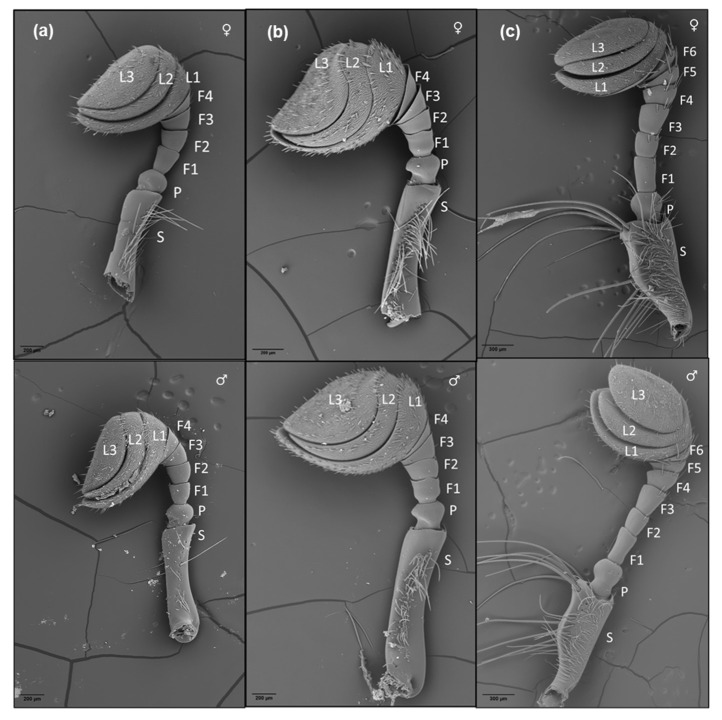
Overview of the general morphology of the male and female antenna of (**a**) *B. bison*, (**b**) *O. aygulus* (scale bars = 200 µm) and (**c**) *G. spiniger* (Scale bars = 300 µm). S: scape, P: pedicel, F1–F6: flagellomeres of the funicle, L1–L3: lamellae of the antennal club.

**Figure 2 insects-14-00627-f002:**
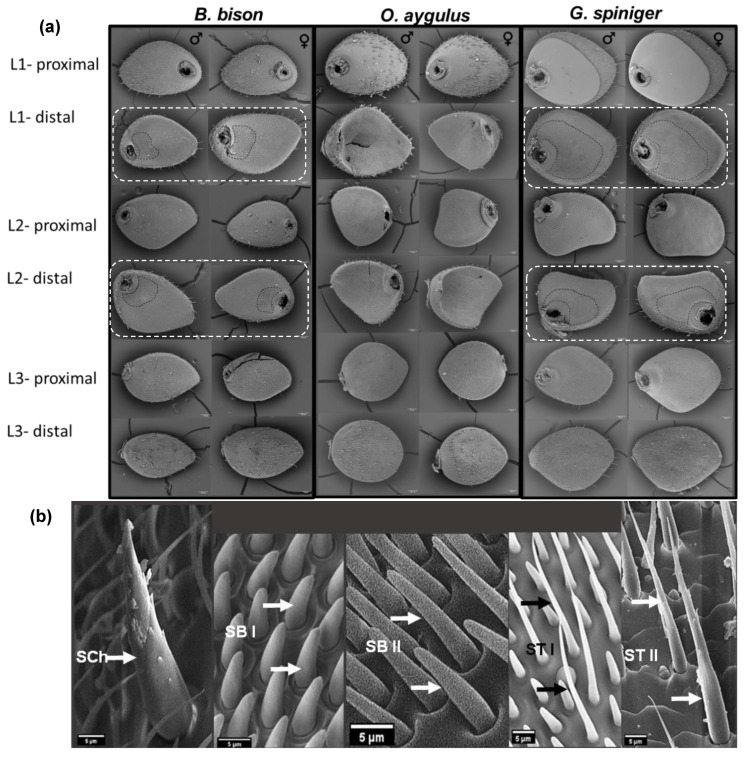
(**a**) SEM images of the three lamellae in the antennae of male and female *Bubas bison*, *Onitis aygulus* and *Geotrupes spiniger* showing proximal and distal surfaces of L1–L3. Distal surfaces of *B. bison* and *G. spiniger* with similar sensilla arrangements are enclosed with dash lines. Scale bars = 100 µm. (**b**) SEM images representing main types of sensilla occurring on the antennal club of dung beetles. From left to right, sensilla chaetica (SCh), sensilla basiconica sub type I (SB I), sensilla basiconica sub type II (SB II), sensilla trichodea sub type I (ST I) and sensilla trichodea sub type II (ST II). Scale bars = 5 µm.

**Figure 3 insects-14-00627-f003:**
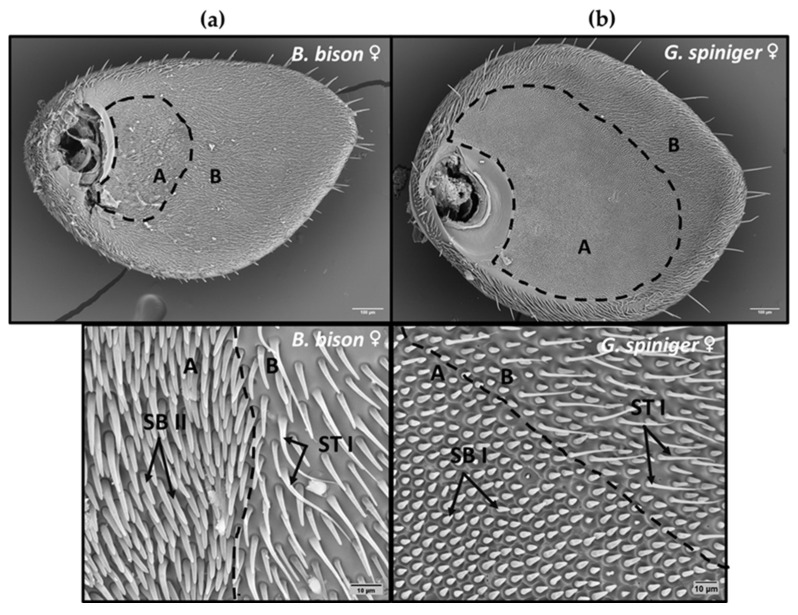
SEM images of L1 distal surface of (**a**) *B. bison* and (**b**) *G. spiniger* showing two zones: **A**—homogenous area (enclosed by a dashed line) and **B**—heterogenous area. Scales bars are 100 µm and 10 µm. Homogenous area of *G. spiniger* consisted of SB I and in *B. bison* it is SB II. Sensilla basiconica sub type I (SB I), sensilla basiconica sub type II (SB II), and sensilla trichodea sub type I (ST I), Scale bars = 10 µm.

**Figure 4 insects-14-00627-f004:**
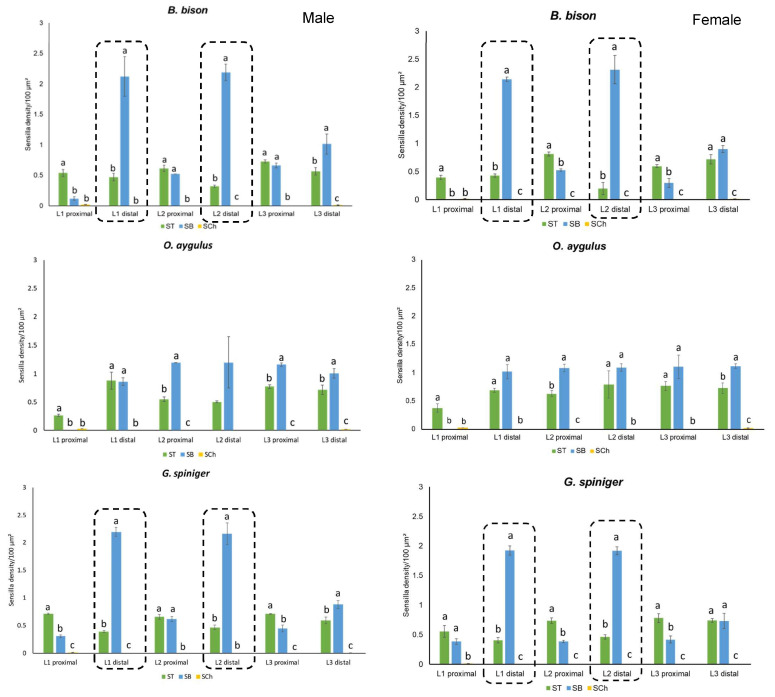
Density and distribution of sensilla trichodea (ST), sensilla basiconica (SB) and sensilla chaetica (SCh) in different lamella surfaces. Bars with different lowercase letters are significantly different at *p* < 0.05 (ANOVA followed by Tukey comparison). In *B. bison* and *G. spiniger,* ST and SB show a similar distribution as indicated by dash lines.

**Figure 5 insects-14-00627-f005:**
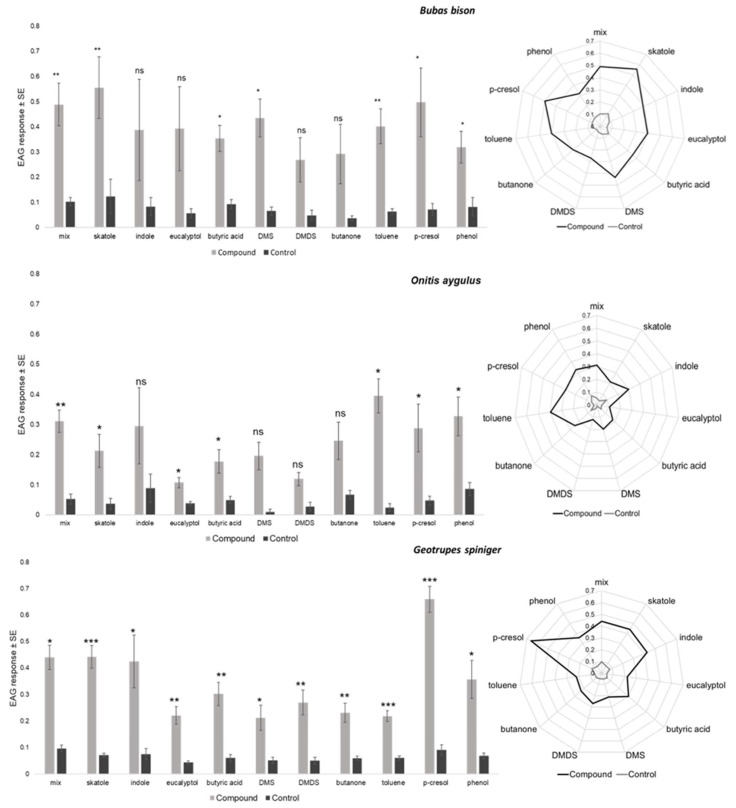
EAG responses (mV) of female *Bubas bison*, *Onitis aygulus* and *Geotrupes spiniger* to selected dung VOCs compared with the subsequesnt control puffs determined by paired *t*-test. * *p* < 0.05, ** *p* < 0.01, *** *p* < 0.001, ns—not significant.

**Figure 6 insects-14-00627-f006:**
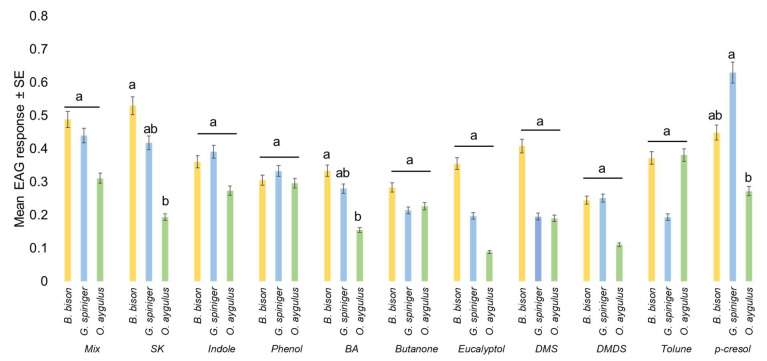
Mean EAG responses (mV) of female *Bubas bison*, *Geotrupes spiniger* and *Onitis aygulus* to dung VOCs compared across three species analysed via ANOVA (*p* < 0.05) followed by LSD. Bars with different letters indicate the significant difference between species using ANOVA and LSD.

**Figure 7 insects-14-00627-f007:**
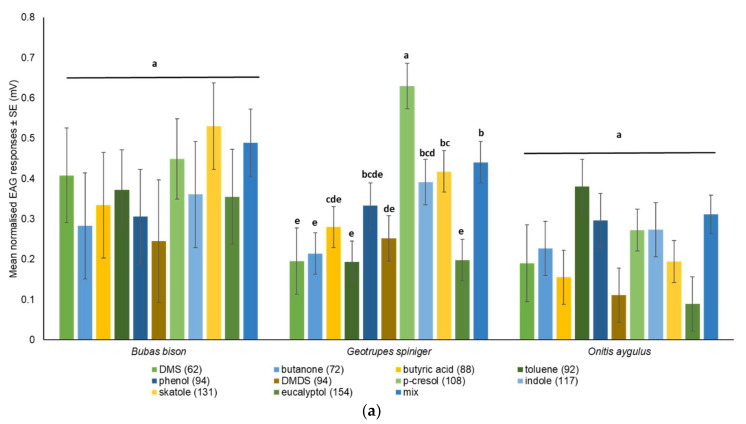
(**a**) Bar chart (**b**) radar chart showing the differences in mean EAG responses (mV) within each species. Significance from ANOVA followed by Fisher’s LSD (for *B. bison* and *G. spiniger*) and Tukey’s pairwise comparisons (*O. aygulus*) are mentioned. Bars with different letters indicate the significant difference between compounds within a species.

**Table 2 insects-14-00627-t002:** Density of various types of sensilla trichodea (ST), sensilla basiconica (SB) and sensilla chaetica (SCh) on both sides of each lamella analysed across lamella surfaces. Different letters in the same column indicate statistical comparison across lamella surfaces defined by ANOVA followed by Tukey’s test (*p* < 0.05). * *p* < 0.05, ** *p* < 0.01 indicate the statistical differences between male and female conspecifics analysed by one-way ANOVA.

Species	Antennal Section	Sensilla Density/100 µm² (Mean ± SE)					
		Sensilla Trichodea		Sensilla Basiconica		Sensilla Chaetica	
		Male	Female	Male	Female	Male	Female
*B. bison*	L1 proximal	0.54 ± 0.049 ^abc^	0.4693 ± 0.05 ^b^	0.12 ± 0.16 ^c^ *	^d^	0.02 ± 0.001 ^a^	0.02 ± 0.002 ^a^
	L1 distal	0.47 ± 0.05 ^bc^	0.43 ± 0.05 ^b^	2.12 ± 0.16 ^a^	2.14 ± 0.08 ^a^	^c^	^c^
	L2 proximal	0.62 ± 0.06 ^ab^	0.81 ± 0.05 ^a^	0.52 ± 0.20 ^bc^	0.53 ± 0.77 ^bc^	^c^	^c^
	L2 distal	0.32 ± 0.05 ^c^	0.43 ± 0.05 ^b^	2.19 ± 0.16 ^a^	2.31 ± 0.09 ^a^	^c^	^c^
	L3 proximal	0.73 ± 0.05 ^a^ *	0.59 ± 0.06 ^ab^	0.66 ± 0.16 ^bc^ *	0.30 ± 0.09 ^cd^	^c^	^c^
	L3 distal	0.57 ± 0.05 ^ab^	0.71 ± 0.47 ^a^	1.01 ± 0.16 ^b^	0.90 ± 0.08 ^b^	0.01 ± 0.002 ^b^	0.016 ± ^b^
*O. aygulus*	L1 proximal	0.27 ± 0.05 ^c^	0.38 ± 0.06 ^cd^	-	0.00 ± 0.11 ^b^	0.03 ± 2.45 × 10^−18 a^	0.03 ± 2.45 × 10^−18 a^
	L1 distal	0.88 ± 0.07 ^a^	0.43 ± 0.06 ^bcd^	0.86 ± 0.16 ^a^	1.02 ± 0.11 ^a^	^c^	^c^
	L2 proximal	0.55 ± 0.05 ^b^	0.81 ± 0.06 ^a^	1.19 ± 0.13 ^a^	1.05 ± 0.11 ^a^	^c^	^c^
	L2 distal	0.50 ± 0.07 ^bc^	0.20 ± 0.08 ^d^	1.20 ± 0.16 ^a^	1.16 ± 0.14 ^a^	^c^	^c^
	L3 proximal	0.77 ± 0.05 ^ab^ *	0.59 ± 0.08 ^abc^	1.16 ± 0.13 ^a^ *	1.11 ± 0.11 ^a^	^c^	^c^
	L3 distal	0.72 ± 0.05 ^ab^	0.71 ± 0.06 ^ab^	1.01 ± 0.13 ^a^	1.12 ± 0.14 ^a^	0.02 ± 0.003 ^b^	0.02 ± 0.002 ^b^
*G. spiniger*	L1 proximal	0.72 ± 0.01 ^a^	0.55 ± 0.10 ^abc^	0.31 ± 0.02 ^c^	0.38 ± 0.04 ^c^	0.01 ± 0.003 ^a^	0.01 ± 0.003 ^a^
	L1 distal	0.39 ± 0.02 ^c^	0.40 ± 0.05 ^c^	2.20 ± 0.08 ^a^	1.92 ± 0.08 ^a^	^b^	^b^
	L2 proximal	0.66 ± 0.05 ^a^	0.74 ± 0.05 ^ab^	0.62 ± 0.04 ^bc^ **	0.39 ± 0.02 ^bc^	^b^	^b^
	L2 distal	0.47 ± 0.04 ^bc^	0.46 ± 0.04 ^bc^	2.16 ± 0.20 ^a^	1.92 ± 0.07 ^a^	^b^	^b^
	L3 proximal	0.71 ± 0.01 ^a^	0.78 ± 0.08 ^a^	0.45 ± 0.06 ^bc^	0.42 ± 0.06 ^bc^	^b^	^b^
	L3 distal	0.59 ± 0.06 ^ab^	0.74 ± 0.03 ^ab^	0.88 ± 0.07 ^b^	0.73 ± 0.13 ^b^	0.01 ± 0.0013 ^ab^	0.005 ± 0.0003 ^b^

**Table 3 insects-14-00627-t003:** Factorial analysis results for the effect of sensilla type, sex, lamella surface and species on the sensilla density. *R*^2^ = 0.968 represents a goodness of fit of the model.

Source of Variation	*df*	*F*	*p*
Sex	1	1.124	0.229
Lamella surface	5	122.323	<0.001
Species	2	1.149	0.319
Sensilla type	2	1419.180	<0.001
Sex × lamella surface	5	0.523	0.759
Sex × species	2	1.311	0.272
Sex × sensilla type	2	3.539	0.031
Lamella surface × species	10	16.057	<0.001
Lamella surface × sensilla type	10	145.910	<0.001
Species × sensilla type	4	8.402	<0.001
Sex × lamella surface × species	10	0.808	0.621
Sex × lamella surface × sensilla type	10	0.798	0.631
Sex × species × sensilla type	4	0.803	0.525
Lamella surface × species × sensilla type	20	27.505	<0.001
Sex × lamella surface × species × sensilla type	20	1.272	0.202

**Table 4 insects-14-00627-t004:** The effect of compounds and species and their interactions of these variables on EAG responses (*R* squared = 0.337).

Source of Variation	*df*	*F*	*p*
Compound	10	2.724	0.005
Species	2	6.615	0.002
Compounds × species	20	0.942	0.536

**Table 5 insects-14-00627-t005:** Pooled EAG responses (mV) compared among the species. Significance from ANOVA followed by Fisher’s LSD (*p* < 0.05). Different letters indicate the significant difference between species.

ANOVA		LSD		
*F*	*p*P	*Bubas bison*	*Geotrupes spiniger*	*Onitis aygulus*
9.37	0.0002	0.397 ± 0.026 ^A^	0.3171 ± 0.025 ^B^	0.2352 ± 0.032 ^C^

**Table 6 insects-14-00627-t006:** Pooled EAG responses (mV) of adult female combined across *B. bison*, *G. spiniger* and *O. aygulus* compared among individual compounds. Significance testing was conducted using ANOVA followed by Fisher’s LSD (*p* < 0.05); compounds sharing the same superscripted letter are not statistically significant.

ANOVA	
*F*	*p*
2.61	0.0059
LSD	
*p*-cresol	0.450 ± 0.048 ^A^
mix	0.430 ± 0.042 ^AB^
skatole	0.391 ± 0.048 ^ABC^
indole	0.352 ± 0.057 ^ABCD^
phenol	0.314 ± 0.055 ^ABCD^
toluene	0.307 ± 0.050 ^BCD^
DMS	0.276 ± 0.055 ^CD^
butyric acid	0.268 ± 0.055 ^CD^
butanone	0.238 ± 0.055 ^D^
eucalyptol	0.231 ± 0.053 ^D^
DMDS	0.211 ± 0.060 ^D^

## Data Availability

All data are stored in archived datasets as per the guidelines of Charles Sturt University and associated funding bodies.

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
