# Peer review of "Characterisation of Antennal Sensilla and Electroantennography Responses of the Dung Beetles Bubas bison, Onitis aygulus and Geotrupes spiniger (Coleoptera: Scarabaeoidea) to Dung Volatile Organic Compounds"

_insects, 2023, doi:10.3390/insects14070627_

Round 1

Reviewer 1 Report

Perera et al. presented original research on dung beetles olfactory sensilla and electroantennography responses from three species in Australia. Authors characterized by scanning electron microscopy the morphology of the antennae and their receptors in detail and compared between males and females as well as across species. They also tested the olfactory responses of females to 10 compounds previously identified to be emanating from dung headspace. Authors confirmed that the three species of dung beetle can smell these compounds and they found some significant differences in sensitivity to each compound among the three species, which could be corelated to their differences in olfactory sensilla identified on their antennae.

Overall, the paper is well presented with sounded methods and statistical analysis.

I have few minor comments:

Authors tend to overinterpret their results from EAG into behavioural attractivity. Even though the odours are coming from dungs which are the main food sources of those beetles, the response of EAG does not preclude that those compounds will be attractive but can be as well repellent. Can authors temper their statement, such as : L49- a chemical lure, if based on repellent odours, won’t be used for trapping; L 145 : use EAG-active compounds as “lure” not attractant

L43: can you explain the choice to test those six additional compounds as well?

L 62: please add reference

L73 : please provide references about dung beetle not other insects as you mentioned “in dung beetles..”

L85-87: I don’t get the point of studying antennal responses if it is not to understand the behaviour

L 87-89: please provide reference

L 97: replace “to exist” by “to signal”

In my uploaded pdf, they were lots of errors for references which could not be found- please address.

Table 1. italicised species names

For SEM, could you mention if you took pictures from the left or right antennae? For some insects, it is known that there is a lateralization of ORs. have you checked it?

For EAG, I was surprised to see that the compounds were diluted in water and not in a solvent like hexan- can you justify.

Figure 1, can you explain why some images are circled? Not explained in the legend.

For all EAG figures and summary tables, please mentioned the unit for the EAG response, such as in mV?

Figure 7. precise in legend that it was normalised mean EAG

L 440: can you explain how you show an association between the antennal sensilla and the EAG response? Is it what you discuss afterwards.

Reviewer 2 Report

This manuscript accumulates enough merits to be published in Insects. First, the authors described for the first time the gross morphology of the male and female antennae of three dung beetle species of economic interest G. spiniger, B. bison and O. aygulus. Secondly, they studied the activity of different volatiles related to HS of dung by electroanthenography and found electrophysiological responses both in pure modality and in mixture, having the corresponding + and - controls. The approach to address their goal is appropriate, the information collected is reliable and the results were discussed appropriately. The information obtained, along with complementary behavioral assays would be useful to build & refine attraction devices. Overall, the manuscript is well-organized and written.

Therefore, this reviewer considers that the manuscript can be published in the present form, asking authors to consider the following points on the proofs stage.
Add all missed references.

_Most of the Figures and Tables are not mentioned in the text.

_Legend of some superscript letters on Tables/Figures are needed.

Line 188, should be Table S5

Line 188, the composition of the mixture.

Line 189, if 10 ug/mL was set to be the optimal concentration by preliminary experiments.

Line 207, a comment that mentions that all tested compounds are assumed to interact with paper and desorbed to headspace equally.

Line 540, correct “of by toluene”.

Line 563, “Characterisation”
